# Porcine Placental Extract Improves the Lipid Profile and Body Weight in a Post-Menopausal Rat Model Without Affecting Reproductive Tissues

**DOI:** 10.3390/nu17060984

**Published:** 2025-03-11

**Authors:** Tugsjargal Purevdorj, Moeka Arata, Mari Nii, Shota Yamamoto, Hiroki Noguchi, Asuka Takeda, Hidenori Aoki, Hiroaki Inui, Tomohiro Kagawa, Riyo Kinouchi, Yuri Yamamoto, Kanako Yoshida, Takeshi Iwasa

**Affiliations:** 1Department of Obstetrics and Gynecology, Institute of Biomedical Sciences, Tokushima University Graduate School, 3-18-15 Kuramoto-cho, Tokushima 770-8503, Japan; tugsjargal.p1216@gmail.com (T.P.); noguchi.hiroki@tokushima-u.ac.jp (H.N.);; 2Department of Renal and Genitourinary Surgery, Graduate School of Medicine, Hokkaido University, Sapporo 060-0808, Japan

**Keywords:** placental extract, weight loss, menopause, ovariectomized rats, alternative therapy, lipid profile

## Abstract

**Introduction:** What if porcine placental extract (PPE) could combat post-menopausal weight gain and lipid imbalances without the side effects of traditional hormone treatments? The menopause marks a critical shift in women’s health, with declining estrogen levels driving increased risks of obesity, metabolic dysfunction, and cardiovascular disease. While hormone replacement therapy remains a common intervention, concerns over its long-term safety have intensified the search for safer alternatives. **Objectives:** This study aims to explore the metabolic effects of porcine placental extract (PPE) by using an ovariectomized (OVX) rat model to mimic the hormonal landscape of the menopause. **Methods:** Twenty OVX Wistar rats were assigned to either a control group receiving phosphate-buffered saline or a PPE-treated group given intraperitoneal PPE injections for two weeks. **Results:** Remarkably, the PPE-treated rats showed significantly lower body weights than the controls. Biochemical analysis revealed that the PPE-treated rats had improved lipid profiles, involving lower total cholesterol and triglyceride levels. Histological examinations of the PPE-treated rats showed no adverse changes in the uterus or mammary glands. **Conclusions:** These results highlight PPE’s potential as a non-hormonal, tissue-safe intervention for combating weight gain and lipid imbalances in post-menopausal conditions. By promoting lipolysis without impacting reproductive health or muscle mass, PPE opens the door to new possibilities for managing post-menopausal metabolic health. However, further research is needed to determine its long-term efficacy.

## 1. Introduction

The menopause marks the end of a woman’s reproductive years, leading to a sharp decline in estrogen that increases the risk of metabolic disorders, cardiovascular disease, obesity, and hormone-sensitive cancers [1,2]. By 2050, over 1.65 billion women will be aged 50 and above, highlighting the urgent need for effective interventions to manage post-menopausal health risks [3]. Metabolic syndrome (MetS) is among the most prevalent concerns in this population, with global prevalence rates ranging from 13.8% to 60%, depending on genetics, lifestyle, and healthcare access [4,5,6]. These variations emphasize the need for a comprehensive approach to managing post-menopausal metabolic health.

Estrogen plays a crucial role in regulating fat distribution, energy metabolism, and lipid balance. Its deficiency after the menopause contributes to increased visceral fat, insulin resistance, and dyslipidemia, elevating the risk of cardiovascular disease and breast cancer [7,8]. While hormone replacement therapy (HRT) has been the standard intervention for managing menopausal symptoms, concerns over its long-term safety, particularly its association with hormone-sensitive cancers, have driven interest in alternative therapies [9,10].

Placental extracts (PPE), rich in bioactive compounds such as peptides, growth factors, and cytokines, have shown potential in promoting lipid metabolism, reducing oxidative stress, and enhancing tissue regeneration [11,12,13]. However, their effects in post-menopausal models remain underexplored.

Ovariectomized (OVX) rat models are widely used to study the physiological and metabolic changes associated with the menopause. These models mimic the hormonal profile of post-menopausal women, making them ideal for evaluating potential therapeutic interventions [14,15]. Previous research on OVX rats has demonstrated that treatments such as oxytocin (OT) can reduce food intake (FI), body weight (BW), and visceral fat, providing insights into possible mechanisms for managing post-menopausal obesity [16,17].

The main objective of this study was to assess changes in lipid profile and body weight following PPE treatment in an OVX rat model. The secondary objectives were to evaluate potential side effects of PPE on reproductive tissues, liver, and kidney function, to determine its safety profile as a non-hormonal intervention. By addressing this gap in the literature, our study contributes to the growing body of research on alternative therapies for managing post-menopausal metabolic health.

## 2. Materials and Methods

### 2.1. Animals

All animal experiments were conducted with the approval of the Ethics Review Committee for Animal Experimentation at Tokushima University. Eight-week-old Wistar female rats (210–240 g) were purchased from Charles River Laboratories Japan, Inc. (Kanagawa city, Japan), and maintained in a room under controlled light (12 h light, 12 h darkness; lights turned on at 08:00 and turned off at 20:00) and temperature (24 °C) conditions, with food and water provided ad libitum.

After a 7-day acclimation period, bilateral ovariectomy was performed on twenty 9-week-old adult rats under sevoflurane-induced anesthesia. Following the surgery, the rats were weighed and housed individually. This study was conducted in accordance with the ARRIVE guidelines for reporting animal research [18].

### 2.2. Porcine Placental Extract and Animal Experiment Protocol

In this study, porcine placental pure powder (Lot. TF3370), obtained from Japan Bio Products Co., Ltd. (Tokyo, Japan), was utilized. Table 1 presents the detailed composition of the powder. The powder was dissolved in phosphate-buffered saline (PBS, pH 7.4) and thoroughly homogenized. To remove any insoluble residues, the solution was subjected to centrifugation at 18,000× *g* for 3 min at room temperature. The resulting extract was adjusted to a final concentration of 60 mg/mL. All preparations were freshly made immediately prior to use to ensure stability and consistency.

Five weeks postoperatively, the BW of the rats was measured. Then, the rats were randomly assigned to one of two groups: (1) the control group (Ctrl group; *n* = 10), which received an intraperitoneal injection of 1 mL of 0.01 mol/L PBS for two weeks, and (2) the PPE group (*n* = 10), which was treated with 200 mg/kg/day of PPE via intraperitoneal injection for two weeks. During the injection period, BW was recorded, and the amount of remaining food was measured to assess FI every day.

Following a 14-day treatment period, the rats were euthanized via decapitation under anesthesia after their BW and remaining food had been measured. Subsequently, biological samples, including part of the brain, visceral fat (the parametrial, perirenal, and mesenteric deposits), subcutaneous fat (the inguinal deposits), mammary glands (left abdominal), uterus, and blood were collected. The wet weights of visceral fat, subcutaneous fat, and the uterus were measured immediately after removal, and tissue samples (around 300–400 mm^3^) of visceral (parametrial) and subcutaneous fat were dissected. Serum was separated out from whole blood by centrifugation and stored at −20 °C, and tissue samples were stored at −80 °C. Other tissue samples of visceral and subcutaneous fat, the mammary gland, and the uterus were fixed in 4% paraformaldehyde.

### 2.3. Biochemical Assay

Whole blood samples were centrifuged at 3000 rpm for 20 min at 4 °C to separate out the serum. The serum was subsequently analyzed at a commercial laboratory (ASKA Pharma Medical Co., Ltd., Kanagawa city, Japan) to determine its OT and leptin levels using a chemiluminescent enzyme immunoassay. Serum was also sent to another commercial laboratory (Oriental Yeast., Co., Itabashi Ward, Tokyo, Japan) for the measurement of serum total protein (TP), albumin, total cholesterol (T-CHO), triglycerides (TG), glucose, high-density lipoprotein (HDL-C), low-density lipoprotein (LDL-C), total bilirubin (T-Bil), creatinine (CRE), and blood urea nitrogen (BUN).

### 2.4. Quantitative Real-Time Polymerase Chain Reaction

Whole hypothalamic explants were dissected from frozen brains, as described previously [19]. Total RNA was extracted from the hypothalamic explants and visceral adipose tissue using the TRIzol^®^ reagent kit (Invitrogen Co., Carlsbad, CA, USA) and the RNeasy^®^ mini kit (Qiagen GmbH, Hilden, Germany). Complementary DNA (cDNA) was synthesized using oligo (deoxythymidine) primers at 50 °C with the SuperScript III first-strand synthesis system (Invitrogen Co.). The real-time polymerase chain reaction (PCR) was performed using the StepOnePlus™ real-time PCR system (PE Applied Biosystems, Foster City, CA, USA) with Fast SYBR^®^ Green. The mRNA expression levels of neuropeptide Y (NPY), pro-opiomelanocortin (POMC), and OT in the hypothalamus, as well as leptin and lipase A (Lipa A) in visceral adipose tissue, were quantified. The expression level of each target gene was normalized to the GAPDH or 18S rRNA level. Dissociation curve analysis was also performed for each gene at the end of the PCR. Each amplicon generated a single peak. The PCR conditions were as follows: initial denaturation and enzyme activation were performed at 95 °C for 20 s, followed by 45 cycles of denaturation at 95 °C for 3 s, and annealing and extension for 30 s.

### 2.5. Histology

Visceral and subcutaneous white adipose tissue, mammary glands, and uterine tissue fixed in formaldehyde were dehydrated using graded series of ethanol and xylene, embedded in paraffin, and sectioned. Serial sections (thickness: 4 μm) were stained with hematoxylin and eosin (H&E). Histological images were captured using a Zeiss Imager M2 microscope equipped with the AxioVision software (version 4.8, Zeiss). Adipose tissue: the mean adipocyte area was quantified by analyzing 100 randomly selected adipocytes per specimen using the OLYMPUS cellSens Standard software, 3.1.1.00. Uterine tissue: horizontal sections of the uterine endometrium encompassing the luminal and glandular epithelia and the lamina propria were examined under ×4 and ×20 magnification to perform detailed histological evaluations. Mammary gland: The stroma acts as the supporting framework of the breast parenchyma. The stroma has fibrous and fatty areas. The parenchyma of the mammary gland is the glandular tissue that produces, stores, and secretes milk. The parenchyma of each mammary is composed of 15 to 20 radially oriented secretory lobules. A lobule comprises glandular tissue, which produces milk; the lactiferous sinus, where the milk is stored; and the lactiferous duct. The parenchymal area per 200 μm of the mammary gland total surface area was measured manually. In addition, the height of the epithelial cells lining the ducts was recorded.

### 2.6. Statistical Analysis

The data are presented as the mean ± standard error of the mean (SEM). Statistical analyses were conducted using the GraphPad Prism 9.5.0 software. The significance of differences was evaluated using two-way repeated 
measures analysis of variance (ANOVA) with the Sidak multiple comparisons test for comparisons among multiple groups. Student’s *t*-test was used for comparisons between two groups. In the figures, asterisks denote statistical significance (*, *p* < 0.05; **, *p* < 0.01; ***, *p* < 0.001; ****, *p* < 0.0001).

## 3. Results

### 3.1. Effects of PPE on Body Weight and Food Intake

The percentage change in BW was significantly lower in the PPE group (324.7 ± 6.04 g) than in the Ctrl group (336.3 ± 6.81 g; *p* < 0.0001, two-way ANOVA, Šídák’s multiple comparisons test; Figure 1A). Similarly, cumulative FI was significantly lower in the PPE group (212.46 ± 5.1 g) than in the Ctrl group (233.43 ± 9.12 g; *p* < 0.0001, two-way ANOVA, Šídák’s multiple comparisons test; Figure 1B).

### 3.2. Effects of PPE on Visceral Fat Weight, Subcutaneous Fat Weight, and Size

Visceral fat weight (g/100 g BW) was significantly lower in the PPE group than in the Ctrl group (*p* < 0.01; df = 18, t = 2.976, Student’s *t*-test). In addition, subcutaneous fat weight (g/100 g BW) was significantly lower in the PPE group than in the Ctrl group (*p* < 0.05; df = 18, t = 2.375, Student’s *t*-test) (Figure 1C). In contrast, no significant differences were observed in lean BW (g) between the PPE and Ctrl groups (Figure 1D). The adipose cells in visceral fat were smaller in the PPE group than in the Ctrl group (*p* < 0.001; df = 18, t = 4.007, Student’s *t*-test; Figure 1E,F).

### 3.3. Effects of PPE on Biochemical Parameters

Significant differences in the lipid profile were observed between the PPE and Ctrl groups. T-CHO levels were significantly lower in the PPE group than in the Ctrl group (*p* < 0.05; t(18) = 2.56), and TG levels were also significantly lower in the PPE group (*p* < 0.05; t(18) = 2.14). Moreover, HDL-C levels were markedly lower in the PPE group than in the Ctrl group (*p* < 0.0001; t(18) = 5.07) (Table 2).

No significant differences in serum OT or leptin levels were observed between the groups. Similarly, blood glucose and LDL-C levels did not show significant differences between the PPE-treated group and the control group. Serum alanine aminotransferase and T-Bil levels were significantly lower in the PPE group than in the Ctrl group (*p* < 0.01; t(18) = 2.88 and *p* < 0.05; t(18) = 2.17, respectively). Despite that, serum CRE levels were slightly higher in the PPE group (*p* < 0.05; t(18) = 2.35). TP, aspartate aminotransferase, and BUN levels did not differ significantly between the groups.

### 3.4. Effects of Central and Peripheral Neuromodulators

The hypothalamic mRNA expression levels of OT, NPY, and POMC did not significantly differ between the PPE and Ctrl groups (Figure 2A). Furthermore, no significant difference was observed in the visceral fat tissue mRNA expression level of leptin or Lipa A between the two groups (Figure 2B). 

### 3.5. Effects of PPE on Uterine and Mammary Gland Tissue

Uterus: The wet weight of the uterus did not differ significantly between the PPE and Ctrl groups (Figure 3A). Histomorphological analysis of the uterus revealed no significant differences in endometrial thickness between the groups, with both exhibiting a similar low cuboidal epithelial cell structure (Figure 3A,B).

Mammary gland: Neither the parenchymal area nor the height of the epithelial cells lining the ducts differed between the PPE and Ctrl groups (Figure 3C). The ducts were lined by atrophic, inactive, and low cuboidal epithelial cells in both the PPE and Ctrl groups because the ovaries had been removed (Figure 3D).

## 4. Discussion

Our study demonstrated that PPE effectively reduced appetite and improved lipid metabolism without adverse effects on uterine, breast, liver, or kidney tissues, suggesting a favorable safety profile. These findings indicate the potential of PPE as a non-hormonal alternative for weight management in post-menopausal women, particularly those at risk of metabolic syndrome and cardiovascular diseases [20,21]. This is especially relevant for women who cannot use, or prefer to avoid, hormone replacement therapy [12,13].

However, further research is needed to elucidate the mechanisms of action, optimize dosing regimens, and validate the safety and efficacy through long-term studies and human clinical trials. Investigating its influence on central nervous system pathways regulating appetite and lipid metabolism will also be essential [14,16,17]. If confirmed, PPE could contribute to developing new therapeutic strategies targeting obesity and metabolic disorders associated with the menopause, offering a safer alternative to hormone replacement therapy [1,22,23].

Mechanisms and implications: One of the key mechanisms observed was related to lipolysis and fat metabolism. PPE has been shown to decrease adipocyte size and fat mass, potentially by enhancing lipolysis. This effect may be mediated through the upregulation of hormone-sensitive lipase (HSL) or adipose triglyceride lipase (ATGL), which are key enzymes in lipid breakdown [24]. Additionally, placental extracts are known to contain peptides and growth factors that can activate AMP-activated protein kinase (AMPK), a crucial regulator of energy balance and fat metabolism [25].

Cholesterol and triglyceride reduction: The observed reduction in total cholesterol and triglyceride levels might be attributed to the modulation of lipid metabolism genes or improved hepatic lipid clearance. Some studies suggest that bioactive peptides in placental extracts can enhance bile acid synthesis, thereby increasing cholesterol excretion [26].

Appetite regulation: Interestingly, the PPE-treated group showed a decrease in food intake without changes in leptin expression or the hypothalamic neurotransmitters involved in appetite regulation. Although no significant changes were observed in hypothalamic neurotransmitters or plasma levels of appetite-regulating hormones, including leptin, this suggests that PPE may reduce appetite through other mechanisms independent of leptin or hypothalamic neurotransmitter signaling. Given that changes were observed in appetite but not in leptin or hypothalamic neuropeptides, other pathways may be involved. For instance, gastrointestinal hormones such as ghrelin, cholecystokinin (CCK), or glucagon-like peptide-1 (GLP-1), which play crucial roles in appetite regulation, could be affected by PPE administration [27]. Additionally, PPE might influence gastrointestinal motility or digestion speed, leading to increased satiety. Another possibility is an impact on the gut microbiome, indirectly altering energy harvest or satiety signaling [20,28]. Further studies are needed to investigate these potential mechanisms.

Safety considerations: Despite metabolic improvements, it is essential to consider potential side effects. In this study, no adverse effects on the uterus, breast, or other organs were observed, and there were no significant changes in liver enzymes or blood sugar levels, suggesting a favorable safety profile. However, long-term safety studies are warranted to confirm these findings [3].

Limitations: This study had several limitations, including the relatively short duration of PPE treatment and the use of a single dose. Future studies should explore different dosing regimens and longer treatment periods to better understand the long-term effects of PPE. Additionally, human clinical trials are needed to confirm these findings and evaluate PPE’s safety and efficacy in post-menopausal women.

While ovariectomized rats are widely used to model post-menopausal conditions, they do not fully replicate the complexity of the human menopause, limiting the generalizability of these results. Variations in lifestyle, genetics, and healthcare access can significantly influence metabolic syndrome prevalence and related health outcomes, as observed in diverse populations [1,6]. Therefore, caution should be exercised when extrapolating these findings to humans, and further research is needed in different demographic groups to enhance generalizability [3].

## 5. Conclusions

In conclusion, this study highlights porcine placental extract (PPE) as a potential non-hormonal intervention for managing post-menopausal metabolic health. PPE treatment significantly reduced body weight and visceral fat, improved lipid profiles, and did not exert adverse effects on reproductive tissues, making it a promising alternative to traditional hormone replacement therapy (HRT). Previous research also suggests that PPE may alleviate menopausal symptoms, further supporting its therapeutic potential in menopause management [29].

Our findings align with these observations, emphasizing the potential of PPE as a non-hormonal approach to improving lipid metabolism and reducing fat mass in post-menopausal conditions. However, the exact mechanisms by which PPE influences metabolic pathways remain unclear. Therefore, future research should focus on elucidating these underlying mechanisms, conducting long-term studies to evaluate its sustained efficacy and safety, and determining optimal dosing strategies for potential clinical applications.

This study contributes to the growing body of evidence supporting PPE as a promising non-hormonal intervention for post-menopausal metabolic health. If validated in human trials, PPE could offer a safer and more accessible alternative to HRT, especially for women seeking non-hormonal interventions for metabolic health.

## Figures and Tables

**Figure 1 nutrients-17-00984-f001:**
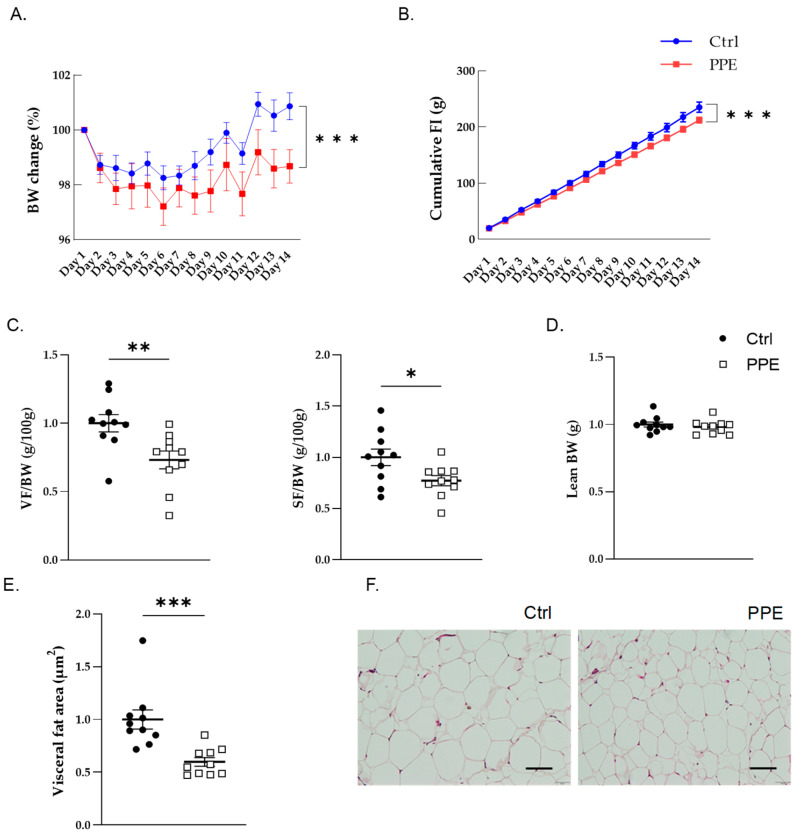
Effects of porcine placental extract (PPE) on body weight (BW), food intake (FI), and body fat. (**A**) BW change. (**B**) Cumulative FI. (**C**) The weight of visceral fat (VF) and subcutaneous fat (SF) per 100 g BW. (**D**) Lean BW. (**E**) Adipocyte size in VF. (**F**) Histological images of the control (Ctrl; blue or black dots) and PPE (red or white squares) groups. Histological tissue samples were stained with hematoxylin-eosin. Magnification: ×20, bar: 20 µm. Data are expressed as the mean ± SEM. * *p* < 0.05, ** *p* < 0.01, *** *p* <0.001.

**Figure 2 nutrients-17-00984-f002:**
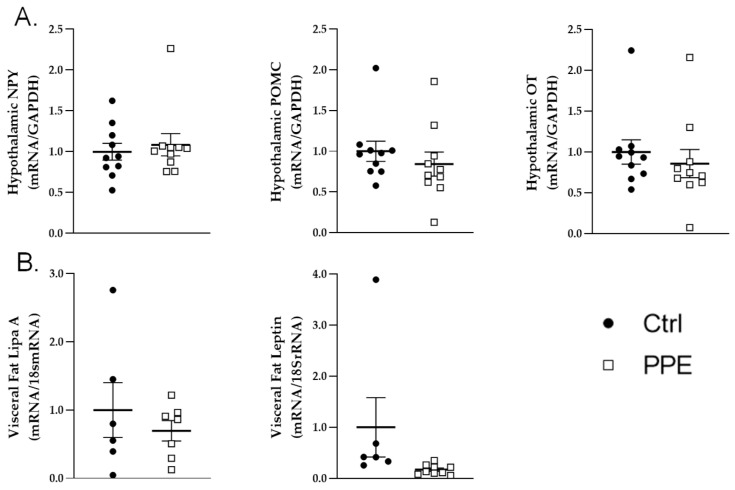
Effects of porcine placental extraction (PPE) on central and peripheral mRNA expression levels. (**A**) Hypothalamic mRNA expression levels of neuropeptide Y (NPY), pro-opiomelanocortin (POMC), and oxytocin (OT) and (**B**) visceral fat mRNA expression levels of lipase A (Lipa A) and leptin in the control (Ctrl; black dots) and PPE (white squares) groups. Data are expressed as the mean ± SEM.

**Figure 3 nutrients-17-00984-f003:**
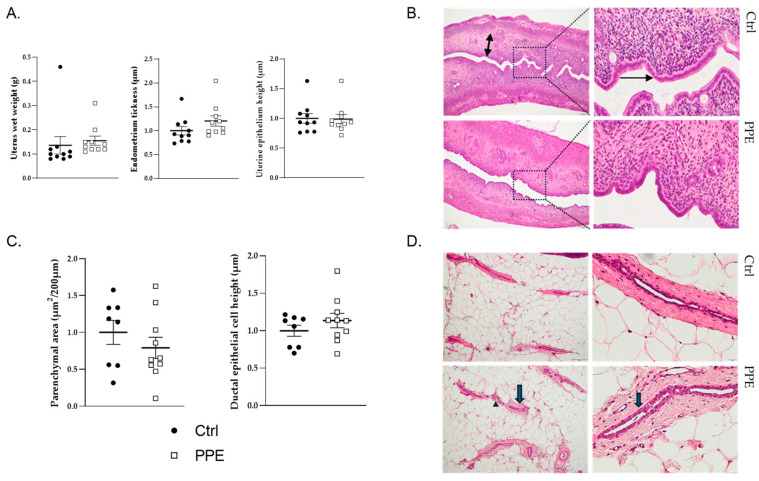
Effects of porcine placental extraction (PPE) on the uterus and mammary gland. (**A**) Uterine weight and endometrial thickness and endometrial epithelium height in the control (Ctrl; ●, black dots) and PPE (□, white squares) groups. (**B**) A representative picture of the endometrial layer (two-sided arrow) and its epithelium (one-sided arrow). (**C**) Mammary gland parenchymal area and the height of the epithelial cells lining the ducts. (**D**) Representative picture of the parenchymal surface containing ducts (arrow) and alveolar (arrowhead) and lobuloalveolar structures (200 µm). Histological tissue samples were stained with hematoxylin-eosin. Left picture: magnification: ×4, bar: 200 µm. Right picture: magnification: ×20, bar: 20 µm.

**Table 1 nutrients-17-00984-t001:** Amino acid analysis of PPE.

Amino Acids (g)	Amount (g/100 g)	Amino Acids (g)	Amount (g/100 g)
Isoleucine (g)	0.92	Tryptophan (g)	0.3
Leucine (g)	1.77	Valine (g)	1.11
Lysine (g)	1.27	Arginine (g)	1.47
Methionine (g)	0.43	Histidine (g)	0.62
Cystine (g)	0.36	Alanine (g)	1.19
Phenylalanine (g)	1.06	Aspartic acid (g)	2.12
Tyrosine (g)	0.74	Proline (g)	1.28
Threonine (g)	0.89	Serine (g)	1.1

**Table 2 nutrients-17-00984-t002:** Effects of porcine placental extract on biochemical parameters.

	Ctrl (*n* = 10)	PPE (*n* = 10)	Reference	*p*-Value
Total Cholesterol (mg/dL)	102.9 ± 3.7	90.1 ± 3.2	81 ± 24	0.01
Triglyceride (mg/dL)	144.2 ± 15.8	97.3 ± 15.0	131.5 ± 76.5	0.04
Glucose (mg/dL)	111.9 ± 4.4	122.9 ± 5.2	202.5 ± 53.5	0.12
HDL-C (mg/dL)	42.9 ± 1.4	34.2 ± 0.8	33.5 ± 8.5	0.00
LDL-C (mg/dL)	9 ± 0.4	9.9 ± 0.5	10 ± 4	0.24
Total Bilirubin (mg/dL)	0.08 ± 0.0	0.068 ± 0.0	0.04 ± 0.01	0.04
Total Protein (g/dL)	6.8 ± 0.1	6.52 ± 0.0	5.9 ± 0.4	0.06
Creatinine (mg/dL)	0.307 ± 0.0	0.338 ± 0.0	0.2 ± 0.05	0.02
BUN (mg/dL)	21.87 ± 0.7	20.96 ± 0.8	20.3 ± 6.1	0.43
AST (IU/L)	180.3 ± 14.2	192.9 ± 12.6	130 ± 36	0.51
ALT (IU/L)	42.4 ± 1.7	35.5 ± 1.6	49 ± 15	0.00
Oxytocin (pg/mL)	21.2 ± 2.4	16.7 ± 1.5	-	0.14
Leptin (ng/mL)	6.8 ± 0.4	6.6 ± 0.6	-	0.74

Ctrl: control group, PPE: porcine placental extract group, HDL-C: high-density lipoprotein cholesterol, LDL-C: low-density lipoprotein cholesterol, BUN: blood urea nitrogen, AST: aspartate aminotransferase, ALT: alanine aminotransferase. Data are expressed as the mean ± SEM.

## Data Availability

The data presented in this study are available upon request from the corresponding author.

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
