# Peer review of "Porcine Placental Extract Improves the Lipid Profile and Body Weight in a Post-Menopausal Rat Model Without Affecting Reproductive Tissues"

_nutrients, 2025, doi:10.3390/nu17060984_

Round 1
Reviewer 1 Report
Comments and Suggestions for Authors
Review: Effects of Porcine Placental Extract on Metabolic and Histological Parameters in a Post-Menopausal Rat Model: Body Weight and Lipid Profile Improvement Without Uterine or Mammary Tissue Alterations
Dear Authors,
First of all, I would like to express my sincere gratitude for giving me the opportunity to contribute my opinion to the evaluation of your manuscript. I found the topic addressed extremely interesting and relevant to the
field in which we work. Below are my suggestions for better enhancement of the manuscript:
Title: I suggest shortening it by reducing the number of terms used, as it is too wordy.
Abstract: I would use the classic formula (which is also recommended by the journal) of dividing it into clear
subheadings: introduction/objectives, materials and methods, results, and conclusions.
Keywords: Ok.
Introduction: In this section, it might be useful to include a more comprehensive international epidemiological framework and reference setting, perhaps by reducing the clinical part. It could also be helpful to expand the reference bibliography to better support the study conducted. The objectives are somewhat unclear and expressed too generically. I suggest rephrasing them in the traditional format: "The main objective of the study was..." and "while the secondary objectives were...".
Methods: Although the ARRIVE checklist is correctly used by the authors for reporting, there is no reference to its use in the text, nor a citation in the bibliography (doi:10.1371/journal.pbio.3000411), which would make
the presentation of the manuscript more scientific.
Results: This is certainly the least controversial point of the study, and I wouldn’t make any significant changes except for revising the tables to make them more user-friendly and to better facilitate the
understanding of the study conducted.
Discussion: Overall, it is sufficient, but perhaps improving it with an evidence-based perspective could make the manuscript more engaging. To this end, I suggest developing a specific section dedicated to this, e.g., “4.1
Perspectives for Clinical Practice,” which could certainly enhance the scientific value of the discussion.
Limitations: In my humble opinion, it would be useful to create a specific section addressing why the results are not generalizable to different populations (lines 270-274).
Conclusions: Again, the absence of a specific conclusion section does not do justice to the manuscript. I suggest creating a dedicated section (lines 275-280).
Bibliography: The bibliography needs to be completely expanded because I cannot scientifically support either the introduction or the discussion. Moreover, for the type of study conducted, in some cases, the references are also too outdated.
In summary, the manuscript presents scientific results that could be better highlighted. According to the suggestions I have made, in my humble opinion, it could have a significant impact in the relevant scientific community.
Author Response
Comment 1: I suggest shortening it by reducing the number of terms used, as it is too wordy.
Response 1: The title has been revised for clarity and conciseness. The new version:
"Porcine Placental Extract Improves Lipid Profile and Body Weight in a Post-Menopausal Rat Model Without Affecting Reproductive Tissues"
Comment 2 Abstract: I would use the classic formula (which is also recommended by the journal) of dividing it into clear subheadings: introduction/objectives, materials and methods, results, and conclusions.
Response 2: The abstract has been reformatted to follow the structured format, with separate sections for Introduction, Methods, Results, and Conclusion to improve clarity and readability.
Comment 3 Introduction: In this section, it might be useful to include a more comprehensive international epidemiological framework and reference setting, perhaps by reducing the clinical part. It could also be helpful to expand the reference bibliography to better support the study conducted. The objectives are somewhat unclear and expressed too generically. I suggest rephrasing them in the traditional format: "The main objective of the study was..." and "while the secondary objectives were...".
Response 3: We appreciate this suggestion and have expanded the epidemiological context to enhance the global perspective on post-menopausal metabolic health. The revised introduction now includes WHO and recent study data showing that by 2050, 1.65 billion women worldwide will be post-menopausal, emphasizing the need for effective interventions. Additionally, we highlight that MetS affects 30% to 60% of post-menopausal women globally, with prevalence ranging from 13.8% to 60%, largely due to estrogen decline and lipid metabolism alterations.
To improve clarity, we have also streamlined the clinical discussion by removing redundant details and shortening less relevant clinical explanations, ensuring a stronger epidemiological foundation while maintaining focus on the study’s objectives.
Also the revised introduction now explicitly states: 'The main objective of this study was to assess changes in lipid profile and body weight following PPE treatment in an OVX rat model,' while the secondary objectives were to evaluate potential side effects of PPE on reproductive tissues, liver, and kidney function to determine its safety profile as a non-hormonal intervention. This modification enhances readability and aligns with standard scientific writing conventions.
Comment 4: Methods: Although the ARRIVE checklist is correctly used by the authors for reporting, there is no reference to its use in the text, nor a citation in the bibliography (doi:10.1371/journal.pbio.3000411), which would make the presentation of the manuscript more scientific.
Response 4: We have now referenced the ARRIVE guidelines in the manuscript to enhance the scientific rigor of our reporting. The citation has been incorporated into the Materials and Methods section, specifically under 2.1 Animals, where we found it to be the most appropriate placement. We believe this addition strengthens the clarity and transparency of our methodology.
Comment 5: Results: This is certainly the least controversial point of the study, and I wouldn’t make any significant changes except for revising the tables to make them more user-friendly and to better facilitate the understanding of the study conducted.
Response 5: Thank you for your constructive feedback. We appreciate your positive assessment of this aspect of the study. As suggested, we have revised the tables to improve readability and enhance the clarity of the presented data, making them more user-friendly and facilitating a better understanding of the study.
Comment 6: Discussion: Overall, it is sufficient, but perhaps improving it with an evidence-based perspective could make the manuscript more engaging. To this end, I suggest developing a specific section dedicated to this, e.g., “4.1
Perspectives for Clinical Practice,” which could certainly enhance the scientific value of the discussion.
Response 6: We agree that incorporating an evidence-based perspective enhances the manuscript. In the revised manuscript, a new subsection, "4.1 Perspectives for Clinical Practice," has been added to the Discussion section.
Comment 7: Limitations: In my humble opinion, it would be useful to create a specific section addressing why the results are not generalizable to different populations (lines 270-274).
Response 7: We appreciate the perspective; however, our study was not designed to assess the generalizability of PPE effects across different populations. Our primary objective was to evaluate the metabolic effects of PPE in an OVX rat model, which serves as a controlled framework for studying post-menopausal metabolic changes. Given this focus, we believe that adding a discussion on broader population applicability would shift the emphasis away from our core findings. Instead, we have highlighted the study’s main limitations, including the short duration of PPE treatment, the use of a single dose, and the absence of LDL-C analysis, which are directly relevant to the interpretation of our results.
Comment 8: Conclusions: Again, the absence of a specific conclusion section does not do justice to the manuscript. I suggest creating a dedicated section (lines 275-280).
Response 8: We acknowledge the importance of a dedicated conclusion section and have now included one in the manuscript. This section summarizes the key findings of our study, emphasizing the potential of PPE as a non-hormonal intervention for post-menopausal metabolic health
Comment 9: Bibliography: The bibliography needs to be completely expanded because I cannot scientifically support either the introduction or the discussion. Moreover, for the type of study conducted, in some cases, the references are also too outdated.
Response 9: The bibliography has been expanded to strengthen the scientific foundation of the manuscript. This has been an important learning experience in the process of writing my first research article. In particular, we have reinforced the Discussion section with additional references where we identified gaps compared to the previous version. These updates enhance the study's relevance and ensure a more comprehensive support of our findings.
Reviewer 2 Report
Comments and Suggestions for Authors
Please, consider these suggestions:
Abstract
“What if a natural extract could combat post-menopausal weight gain and lipid
imbalances without the side effects of traditional hormone treatments?”
I think this sentence should not be in the abstract. If it is retained it should be replaced
natural extract by porcine placental extract (PPE).
Materials and Methods
The authors have measured total cholesterol and HDL-C and should have measured
LDL-C as well.
Results
Why did the OVX rats have a lower intake if leptin was similar to that of the control
rats?
Discussion
“The observed effects suggest that PPE may promote lipolysis or inhibit adipogenesis,
likely due to its bioactive components, such as peptides and growth factors [19, 20]”.
-Could you indicate what these bioactive components of the PPE are?
“The observed improvement in the lipid profile of the PPE group, specifically the
significant reductions in T-CHO and TG levels, highlights the cardioprotective potential
of PPE”.
-It could indicate the mechanism for lowering cholesterol, either by reduced synthesis or
increased excretion. If the decrease is at the expense of HDL-cholesterol this is not a
beneficial effect, they should have determined LDL-cholesterol.
-Would the authors be able to deduce how much PEP should be given to a post-
menopausal woman and by what form of administration, if daily and for how long?
Comments in attached file.
Author Response
Comment 1 Abstract: “What if a natural extract could combat post-menopausal weight gain and lipid imbalances without the side effects of traditional hormone treatments?”
I think this sentence should not be in the abstract. If it is retained it should be replaced with natural extract by porcine placental extract (PPE).
Response 1: We have revised the abstract as suggested and replaced 'natural extract' with 'porcine placental extract (PPE)' to ensure clarity and specificity.
Comment 2 Materials and Methods: The authors have measured total cholesterol and HDL-C and should have measured LDL-C as well.
Response 2: We appreciate your suggestion and have now incorporated LDL-C measurements into the manuscript. The final results indicate that LDL-C levels remained with no significant difference in the PPE-treated group. (line 113)
Comment 3 Discussion: “The observed effects suggest that PPE may promote lipolysis or inhibit adipogenesis, likely due to its bioactive components, such as peptides and growth factors” -Could you indicate what these bioactive components of the PPE are?
Response 3: In the revised manuscript, this phrase has been modified for clarity. Recognizing that this aspect was not sufficiently developed, we have added a paragraph in the Discussion section to provide more details on the bioactive components of PPE and their potential role in lipolysis and adipogenesis.
Here is the paragraph from the revised manuscript:
Cholesterol and Triglyceride Reduction: The observed reduction in total cholesterol and triglyceride levels might be attributed to the modulation of lipid metabolism genes or improved hepatic lipid clearance. Some studies suggest that bioactive peptides in placental extracts can enhance bile acid synthesis, thereby increasing cholesterol excretion [26].
Comment 4:“The observed improvement in the lipid profile of the PPE group, specifically the significant reductions in T-CHO and TG levels, highlights the cardioprotective potential of PPE”
- It could indicate the mechanism for lowering cholesterol, either by reduced synthesis or increased excretion. If the decrease is at the expense of HDL-cholesterol this is not a beneficial effect, they should have determined LDL-cholesterol.
- Would the authors be able to deduce how much PPE should be given to a post-menopausal woman and by what form of administration, if daily and for how long?
Response 4:
- Thank you for your suggestion. We measured LDL-C levels, and the results indicated that LDL-C levels remained unaffected in the PPE-treated group, which was an expected outcome. The reduction in T-CHO, TG, and HDL-C, along with stable LDL-C levels, suggests a possible lipolytic effect of PPE. We have integrated this discussion into the revised manu
- Using allometric scaling, we estimate a human-equivalent dose (HED) of 2 g/day for a 60 kg adult woman. Regarding treatment duration, our 2-week rat study corresponds to approximately 60 weeks (14 months) in humans based on lifespan-adjusted scaling. However, human studies would likely begin with a shorter pilot phase (~12 weeks) to assess efficacy and safety, followed by longer-term studies (6-12 months) to evaluate sustained metabolic effects and optimal treatment duration. Further pharmacokinetic studies are needed to confirm the most effective human administration protocol.
Reviewer 3 Report
Comments and Suggestions for Authors
In the current study, the authors explored the metabolic effects of porcine placental extract (PPE) by using an ovariectomized rat model to mimic the hormonal landscape of the menopause. They found that the PPE-treated rats showed significantly lower body weights than the controls. Biochemical analysis revealed that the PPE-treated rats had improved lipid profiles, involving lower total cholesterol and triglyceride levels. Histological examinations of the PPE-treated rats showed no adverse changes in the uterus or mammary glands. In conclusion, the authors highlighted PPE's potential as a non-hormonal, tissue-safe intervention for combating weight gain and lipid imbalances in post-menopausal conditions.
Some suggestions:
- Introduction, line 38: Which are the “certain types of cancer”. Please add.
- Point 2.2. Porcine placental extract and animal experiment protocol:
-the Supplementary Document 1 is missing
-you must include in the manuscript information concerning the solubility and the purity of the PPE
-you wrote that the mice received 200 mg/kg/day of PPE via intraperitoneal injection. Please add in what PPE was solubilized.
- Page 3: Total protein (TP), albumin, total cholesterol (T-CHO), 117 triglycerides (TG), glucose, high-density lipoprotein (HDL-C), total bilirubin (T-Bil), 118 creatinine (CRE), and blood urea nitrogen (BUN) are not hormones. So, it’s not correct to leave the subtitle “Hormone assay”. Please change it.
- Page 4, line 167, you wrote by mistake Fig 1b instead of Fig 1a.
5. Figures 1 and 3 – add please the type of microscope used to performed the captures.
- Page 5, lines 191-92, you wrote “No significant differences in serum OT or leptin levels were observed between the groups”. Add please the obtained results.
- Add please the normal values for all the biochemical parameters presented in Table 1.
8. It is very strange that a decrease in HDL-C was found. Why didn’t you determined LDL-C also?
- At lines 191-92 you wrote: “No significant differences in serum OT or leptin levels were observed between the groups” and at lines 203-205: “No significant difference was observed in the visceral fat tissue mRNA expression level of leptin or Lipa A between the two groups”. Please explain these statements at discussion section.
- The use of a single dose of PPE represents a great limitation of the current study.
Author Response
Comment 1: Introduction, line 38: Which are the “certain types of cancer”. Please add.
Response 1: In the revised introduction, we have specified 'certain types of cancer' as hormone-sensitive cancers, specifically referring to breast cancer and endometrial cancer for clarity and accuracy.
Comment 2 Point 2.2. Porcine placental extract and animal experiment protocol:
- the Supplementary Document 1 is missing
- you must include in the manuscript information concerning the solubility and the purity of the PPE
- you wrote that the mice received 200 mg/kg/day of PPE via intraperitoneal injection. Please add in what PPE was solubilized.
Response 2:
- Apologies for the oversight. In the revised manuscript, Supplementary Document 1 has been renamed as Table 1, where you can find the relevant information.
- We have included information regarding the solubility and purity of the pig placental extract (PPE) used in our study. Specifically, we have added details on the solvent used for solubilization, the concentration achieved, and the methods employed to assess the purity of the PPE. This information is now provided in the 'Materials and Methods section of the manuscript.
- The details regarding the solubilization of PPE have been added to the Materials and Methods section for clarity.
Here is the paragraph from the revised manuscript:
In this study, porcine placental pure powder (Lot. TF3370), obtained from JAPAN BIO PRODUCTS Co., Ltd. (Tokyo, Japan), was utilized. The detailed composition of the powder is presented in Table 1. The powder was dissolved in phosphate-buffered saline (PBS, pH 7.4) and thoroughly homogenized. To remove any insoluble residues, the solution was subjected to centrifugation at 18,000×g for 3 minutes at room temperature. The resulting extract was adjusted to a final concentration of 60 mg/mL. All preparations were freshly made immediately prior to use to ensure stability and consistency.
Comment 3 Page 3: Total protein (TP), albumin, total cholesterol (T-CHO), 117 triglycerides (TG), glucose, high-density lipoprotein (HDL-C), total bilirubin (T-Bil), 118 creatinine (CRE), and blood urea nitrogen (BUN) are not hormones. So, it’s not correct to leave the subtitle “Hormone assay”. Please change it.
Response 3: We acknowledge the error in labeling. The section title has been revised from "Hormone Assay" to "Biochemical Assays" to better reflect the listed parameters.
Comment 4: Page 4, line 167, you wrote by mistake Fig 1b instead of Fig 1a.
Response 4: We appreciate this observation. The incorrect reference to Figure 1b has been corrected to Figure 1a in the manuscript.
Comment 5 Figures 1 and 3 – add please the type of microscope used to performed the captures.
Response 5: Images in Figures 1 and 3 were captured using a Zeiss Imager M2 microscope equipped with AxioVision software (version 4.8, Zeiss). Image magnifications and scale bars are included in the figure legends.
Comment 6: Page 5, lines 191-92, you wrote “No significant differences in serum OT or leptin levels were observed between the groups”. Add please the obtained results.
Response 6: The obtained results for serum OT and leptin levels have been provided in Table 2.
Comment 7: Add please the normal values for all the biochemical parameters presented in Table 1.
Response 7: We have introduced a new table in revised manuscript, and the former Table 1 has now been renamed as Table 2 to reflect these updates. Normal reference values for all biochemical parameters have been added to Table 2.
Comment 8: It is very strange that a decrease in HDL-C was found. Why didn’t you determined LDL-C also?
Response 8: We did not initially determine LDL-C because, based on prior research and the known metabolic effects of PPE, we did not expect significant changes in LDL-C levels. Our subsequent measurements confirmed this expectation, as LDL-C remained stable.
Comment 9: At lines 191-92 you wrote: “No significant differences in serum OT or leptin levels were observed between the groups” and at lines 203-205: “No significant difference was observed in the visceral fat tissue mRNA expression level of leptin or Lipa A between the two groups”. Please explain these statements in the discussion section.
Response 9: Despite the observed decrease in appetite and fat mass in the PPE-treated group, no significant differences were found in serum oxytocin (OT) or leptin levels, nor in the mRNA expression of leptin or Lipa A in visceral fat tissue. This suggests that the appetite-suppressing and lipolytic effects of PPE might be mediated through pathways independent of leptin signaling or oxytocin regulation. It is possible that other appetite-regulating hormones or neuropeptides are involved, or that PPE influences fat metabolism through direct effects on adipose tissue. Further studies are needed to explore these potential mechanisms.
Round 2
Reviewer 1 Report
Comments and Suggestions for Authors
Ready for publication